# The Efficacy of Photobiomodulation Therapy in Improving Tissue Resilience and Healing of Radiation Skin Damage

Rodrigo Crespo Mosca [1], Sofia Nascimento Santos [2], Gesse Eduardo Calvo Nogueira [3], Daisa Lima Pereira [3], Francielli Campos Costa [4], Jonathas Xavier Pereira [5], Carlos Alberto Zeituni [4] and Praveen Ravindra Arany [1,*]

[1] Oral Biology, Biomedical Engineering and Surgery, School of Dental Medicine, University at Buffalo, 3435 Main Street, Buffalo, NY 14214, USA; rcmosca@alumni.usp.br

[2] Energetic and Nuclear Research Institute (IPEN/CNEN-SP)–CR–Radiation Center, Ave. Lineu Prestes, 2242 Cidade Universitária, São Paulo 05508-000, Brazil; snsantos85@gmail.com

[3] Energetic and Nuclear Research Institute (IPEN/CNEN-SP)–CLA–Center for Laser and Applications, Ave. Lineu Prestes, 2242 Cidade Universitária, São Paulo 05508-000, Brazil; gesse@usp.br (G.E.C.N.); daisalima@gmail.com (D.L.P.)

[4] Energetic and Nuclear Research Institute (IPEN/CNEN-SP)–CTR–Radiation Technology Center, São Paulo 05508-000, Brazil; fran_ccampos@hotmail.com (F.C.C.); czeituni@ipen.br (C.A.Z.)

[5] Institute of Biomedical Sciences, Federal University of Rio de Janeiro, Rio de Janeiro 21941-590, Brazil; jonathasxp@gmail.com

\* Correspondence: prarany@buffalo.edu; Tel.: +1-(716)-829-3479

**Abstract:** The increased precision, efficacy, and safety of radiation brachytherapy has tremendously improved its popularity in cancer care. However, an unfortunate side effect of this therapy involves localized skin damage and breakdown that are managed palliatively currently. This study was motivated by prior reports on the efficacy of photobiomodulation (PBM) therapy in improving tissue resilience and wound healing. We evaluated the efficacy of PBM therapy on 36 athymic mice with $^{125}$I seed (0.42 mCi) implantation over 60 days. PBM treatments were performed with either red (660 nm) or near-infrared (880 nm, NIR) LEDs irradiance of 40 mW/cm$^2$, continuous wave, fluence of 20 J/cm$^2$ once per week. Animals were evaluated every 7 days with digital imaging, laser Doppler flowmetry, thermal imaging, µPET-CT imaging using $^{18}$F-FDG, and histology. We observed that both PBM treatments—red and NIR—demonstrated significantly less incidence and severity and improved healing with skin radionecrosis. Radiation exposed tissues had improved functional parameters such as vascular perfusion, reduced inflammation, and metabolic derangement following PBM therapy. Histological analysis confirmed these observations with minimal damage and resolution in tissues exposed to radiation. To our knowledge, this is the first report on the successful use of PBM therapy for brachytherapy. The results from this study support future mechanistic lab studies and controlled human clinical studies to utilize this innovative therapy in managing side effects from radiation cancer treatments.

**Keywords:** photobiomodulation therapy; brachytherapy; radiation wounds; LED; µPET-CT

## 1. Introduction

Cancer incidence has increased significantly in recent years due to continuous population growth and aging. With the advancement in biology and technologies, current cancer treatment usually consists of individual chemotherapy or combined use of chemotherapy, surgery, radiotherapy, and immunotherapy depending on the etiology of the tumor [1,2]. Radiotherapy is delivered through two methods—an external ionizing radiation beam (teletherapy) or by an implantable internal source (brachytherapy) [3]. In teletherapy, an ionizing radiation beam must be transmitted through upper layers of adjacent healthy tissues before the target tumor cells receive the appropriate radiation dose. This results in normal superficial tissues receiving high radiation doses. Alternatively, brachytherapy

delivers ionizing radiation from sealed metallic cylinders (commonly termed as seeds) containing radioactive isotopes implanted within the cancer tissue. Thus, in brachytherapy, the target (cancer) tissue receives a high dose of radiation, while healthy surrounding tissues are exposed to lower doses, reducing the potential for side effects [4,5]. The two most common radioactive isotopes used in brachytherapy seeds are iodine-125 ([125]I) and palladium-103 ([103]Pd). The shorter half-life (16.96 days) of [103]Pd enables a faster dose rate, compared with [125]I, whose half-life is 59.408 days. Utilizing these differences in dose rates, an isotope is chosen based on specific tumor characteristics; for example, slow-growing, initial tumors are treated with [125]I, while faster-growing, more aggressive tumors are treated with [103]Pd [6].

Radiotherapy complications can range from mild erythema to severe radionecrosis can affect exposed healthy tissues [7,8]. These complications are most marked in tissues containing cells with high metabolic and proliferation activity, such as mucosa (oral and gastrointestinal) and skin (hair follicles) tissues [3]. Current treatments to manage chronic radionecrosis ulcers involve routine wound care principles such as the maintenance of an optimal wound environment to promote granulation, suitable dressing, and topical antimicrobial agents. Pain associated with these ulcerations is a major contributing factor in impacting the quality of life in these patients that are largely managed palliatively with extensive use of systemic anti-inflammatory and analgesic drugs, especially opioids, that can lead to dependence, lethargy, and gastric injuries [9–11]. Surgical interventions are also available for the management of radionecrosis-affected tissues after the acute inflammatory phase involving debridement of necrotic tissue and reconstructive repair [12].

Novel therapies for cutaneous radionecrosis are aimed at revitalizing or aiding the repair of radiation-damaged skin [13]. Among them, the use of low-dose light therapy termed photobiomodulation (PBM; formerly low-level light therapy (LLLT)) has gained much recent attention [14–17]. For over 40 years, PBM has been known to accelerate the healing of acute and chronic wounds [14,18–20]. Among the three categories of PBM mechanisms, the effect of red and near-infrared light to directly modulate the mitochondrial enzymes, cytochrome C oxidase (Complex IV) has been noted [21,22]. The direct consequences of enhanced mitochondrial activity lead to higher ATP/ADP ratios and transitory mild oxidative stress. Both effects activate several signaling cascades (e.g., AP1 and NFk-B pathways) that can induce cellular proliferation, migration, apoptosis inhibition, and intense protein and nucleic acid synthesis. Such cellular effects can lead to modulation of inflammatory processes, impaired pain signaling, and optimized tissue regeneration [20]. The other key PBM mechanism involves the inactivation of photosensitive cell membrane receptors such as TRPV1 and Opsins that mediate analgesia. Finally, direct activation of extracellular latent TGF-β1, a potent wound healing factor, by PBM treatments has been shown to promote tissue healing and repair [23].

A major recent milestone for the PBM field was a recently published systematic review and meta-analysis by the Multinational Association of Supportive Care in Cancer, recommending its use in supportive cancer care in managing oral mucositis [24]. Their analysis showed clear clinical evidence that PBM treatments improved tissue resilience to reduce the incidence and severity of OM following oncotherapy. Similar clinical observations for radiation damage and chronic wounds have been reported [15]. This study was motivated by these observations and inquired if [125]I brachytherapy-induced radionecrosis in the skin in athymic mice could be effectively managed with PBM treatments. To objectively examine the therapeutic responses, a battery of outcomes including clinical wound and thermal imaging, laser Doppler for perfusion, PET-CT for soft tissue metabolic analysis, and histology were assessed over a time course.

## 2. Materials and Methods

### 2.1. Iodine Seeds

Seeds were produced containing $^{125}$I in the Radiation Technology Center (CTR) of the Energetic and Nuclear Research Institute (IPEN/CNEN-SP). The iodine-125 compounds were adsorbed on a silver wire of 0.5 mm diameter and 3 mm in length. Each silver wire was enclosed in the seed, which consisted of a titanium capsule of 0.8 mm in diameter, 0.05 mm wall thickness, and 4.5 mm in length. The seed had a reference activity of 0.4252 mCi and Kerma intensity in the air of 0.931 µGy m$^2$ h$^{-1}$.

### 2.2. Animal Procedures

In total, 36 *Nude* female mice (Nu/Nu), 8 weeks old, weighing 20 $\pm$ 5 g. These mice were kept in sterile cages and acclimatized shelves, at 12/12 h light–dark cycles, and with water and granulated food served ad libitum. The animals were randomly distributed into 6 groups with 6 animals each. Animals were chosen randomly to be allocated to each group to eliminate bias and create homogeneous groups labelled as controls (no interventions), red laser treatment, near-infrared laser treatment, radiation alone, radiation with red laser treatment, and radiation with near-infrared laser treatments. Following institutional ethical approval, all animals were anesthetized using a custom-made isoflurane vaporizer device (*patent pending*) at 3% during the induction phase and maintained by intramuscular injection of 0.3 mL of ketamine and xylazine solution diluted in saline (1.0 mL Ketamine + 0.5 mL xylazine + 8.5 mL saline) prior to surgical insertion of the radioisotope seeds subcutaneously in the infrascapular area on the dorsum skin in mice.

### 2.3. Photobiomodulation (PBM) Therapy

To perform PBM, a LED device with a 1 cm$^2$ beam spot size (Blackbox Mini LEDsabr, Biolambda, Sao Paulo, Brazil, 2019) was used with wavelengths at 660 nm and 880 nm with an irradiance of 40 mW/cm$^2$, continuous wave, fluence of 20 J/cm$^2$, directly in contact with the back of the mice over the site of seed insertion. PBM treatments were started on day 0 of the study and administrated once per week till 60 days at end of the study. These dosing parameters were based on our prior dose-escalation studies for PBM treatments [15].

### 2.4. Wound-Image Analyses

The animals were photographed as soon as the first sign of radionecrosis and every 7 days till healing was complete. Animals were anesthetized using a custom-made isoflurane vaporizer device (*patent pending*) at 3% of concentrations for the induction phase. Images were captured with a standard setup using a digital camera (Nikon, Tokyo, Japan) placed at 5 cm above the animal using a custom stand. The images collected were analyzed using the software, NIH ImageJ to quantify wound area.

### 2.5. Tissue Perfusion Analysis

To assess vascular perfusion in tissues, laser Doppler flow was performed as described previously [25]. Briefly, a Flolab flowmeter with MP13 probe (Moor Instruments Ltd., Axminster, Devon, UK), equipped with a 1 mW laser emitting at 780 nm at 15 Hz, was used 4 cm above the mouse skin. The MP13 is a noncontact probe that avoids flow alterations due to mechanical contact with the mouse skin. The LDF laser output power was validated using a calibrated detector (Laser Check, Coherent, Santa Clara, CA, USA). The LDF output signal is named Flux (F) or Perfusion and denoted in arbitrary units (a.u.). Analysis was performed for 1 min at two selected square sites (2 $\times$ 2 cm each) on the skin at radionecrosis, and a healthy site located immediately caudal to the above site (Figure 1). Animals were acclimatized to room conditions at 22 °C for 10 min prior to assessments, and weekly imaging was performed at the same time of day to reduce metabolic or circadian variations.

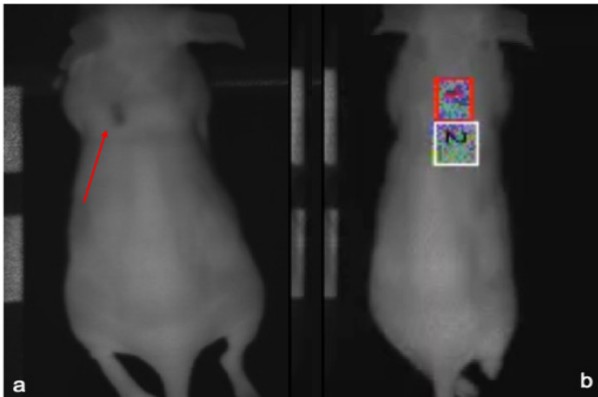

**Figure 1.** High-definition images captured from laser Doppler flow showing in (**a**) red arrow indicated the radionecrosis area and (**b**) assessment areas over the radionecrosis lesion (red square) and normal skin below (white square).

### 2.6. Tissue Temperature to Assess Inflammation

To assess persistent inflammation, skin thermal imaging was performed as described previously [26]. Briefly, a thermal camera (FLIR SC5600, FLIR System Inc., Wilsonville, OR, USA) was fixed at a 4 cm distance from mouse skin, and images were recorded weekly at camera frame rate has 5 Hz and 640 × 480 pixels resolution with emissivity set at 0.98 for 30 s. The detectable temperature range was 5–57 °C with a resolution of 0.5 °C. The temperature readings were performed in two circular areas, 2 cm in diameter over the radionecrosis, and the healthy site immediately caudal to the above site.

### 2.7. Metabolic Analysis with Micro-Positron Emission Tomography (µPET-CT) Imaging

To precisely assess soft tissue changes, we performed µPET-CT imaging (Inveon, Siemens, Knoxville, TN, USA) at 42 days post-radiation exposure. Following complete healing of the radionecrosis lesion in the radiation control group, animals from the two PBM, red and NIR groups, were anesthetized with 3% isoflurane. Then, 50 µL of radioactive fluorodeoxyglucose ($^{18}$F-FDG, activity between 200 and 300 µCi with radiometer) was injected through the caudal vein. After 45 min to allow uptake and biodistribution, image acquisition was performed and analyzed using Amide 1.0.4 (Andreas Loening).

### 2.8. Histology

To examine tissue responses, one representative animal per group was sacrificed at 42 days that demonstrates maximal severity post-radiation. Full-thickness skin biopsies were obtained from each group, and half were immediately fixed in 4% paraformaldehyde and processed routinely for hematoxylin and eosin staining. The other half was fixed in 10% formalin, embedded in paraffin, and processed routinely for Masson's Trichrome staining.

### 2.9. Statistical Analysis

Data were organized in Excel, and statistical analyses were performed using GraphPad Prism 7.0 (GraphPad Software Inc., San Diego, CA, USA). The data from wound area, skin thermal, and tissue perfusion imaging were compared with the untreated control group using Kruskal–Wallis and Dunn's test.

## 3. Results

### 3.1. Radionecrosis Lesions and PBM Treatments

We first examined the effects of $^{125}$I brachytherapy on mouse health and observed no overall systemic or constitutional effects. The total dose was ~$8.5 \times 10^4$ Sv when the first signs of radionecrosis appeared on the skin at 21 days in all groups. Quantitative digital wound analysis following PBM treatments with both NIR and Red LEDs demonstrated significantly reduced incidence and severity of radionecrosis (Figure 2). Interestingly,

treatments with red (660 nm) PBM treatments appeared to be more effective at mitigating skin damage than NIR.

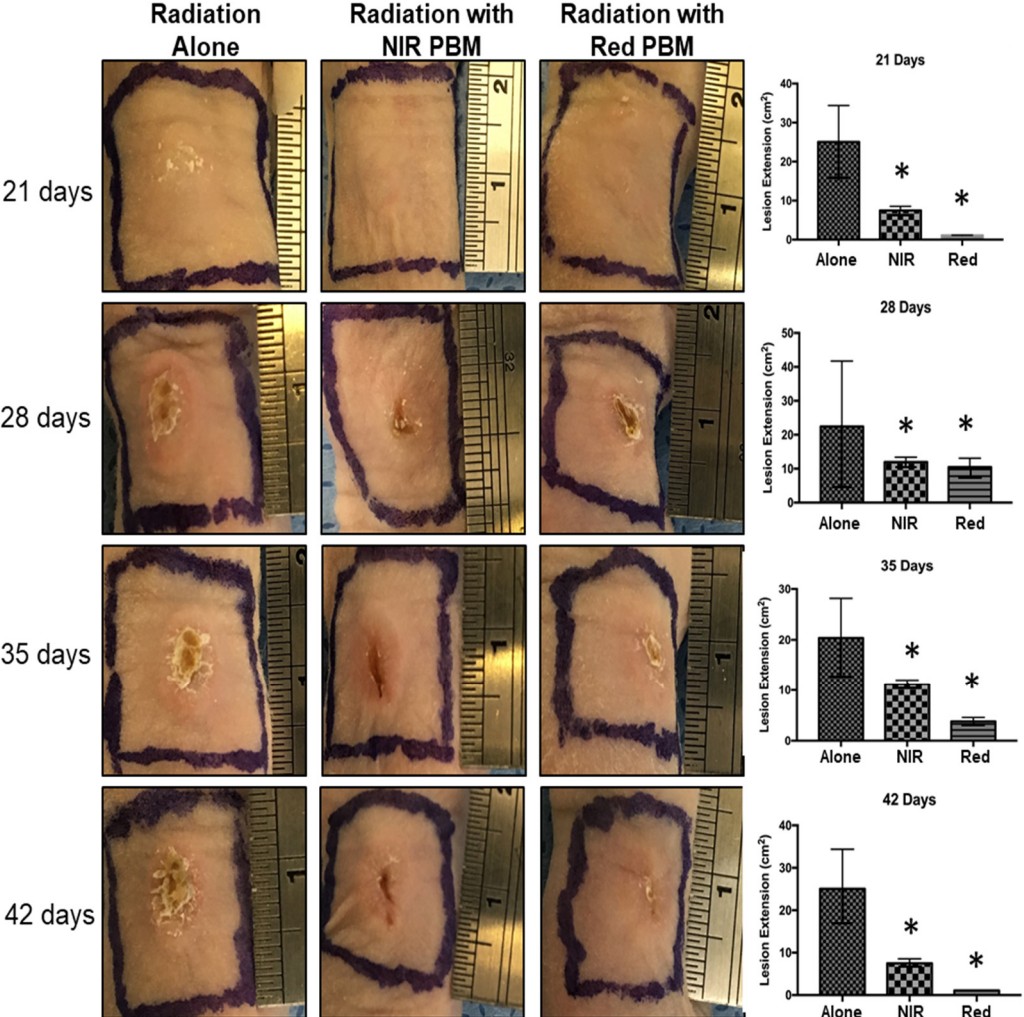

**Figure 2.** Wound images from representative animal groups over time after $^{125}$I brachytherapy and PBM treatments at 21, 28, 35, and 42 days Digital quantitation of lesions is shown on the right panel as means with standard deviation; * denotes statistical significance $n = 6$, $p < 0.05$ with Kruskal–Wallis and Dunn's test.

We also examined the time course of development and resolution of the radionecrosis lesions in all the groups. We noted a delayed onset and reduced severity of radionecrosis lesions with both NIR and Red PBM therapy (Table 1). Moreover, we observed a reduced time to healing with both PBM groups. These responses were more prominent with red PBM treatments compared to NIR treatments.

**Table 1.** Time course of radionecrosis lesion in animal study groups.

| Radionecrosis (Days Post $^{125}$I Seeding) | Radiation Alone Group | Radiation NIR-PBM Group | Radiation Red-PBM Group |
|---|---|---|---|
| First Sign | 21 | 21 | 21 |
| Maximum Severity | 42 | 35 | 28 |
| Healing | 61 | 49 | 42 |

### 3.2. Tissue Perfusion Analysis

The radiation-induced necrotic tissue damage results from prominent vasculitis resulting in a prominent reduction in vascular perfusion. Thus, a good measure of radiation-damaged tissue health is restoration or maintenance of vascular flow. To assess this, skin overlying the implanted $^{125}$I seed was assessed with laser Doppler flowmetry and compared with adjacent normal-appearing skin. We observed improved cutaneous vascular perfusion in the radionecrosis lesions in the PBM-treated groups, compared with radiation treatment alone at 42 days (Figure 3). These data indicate improved skin perfusion following PBM treatments.

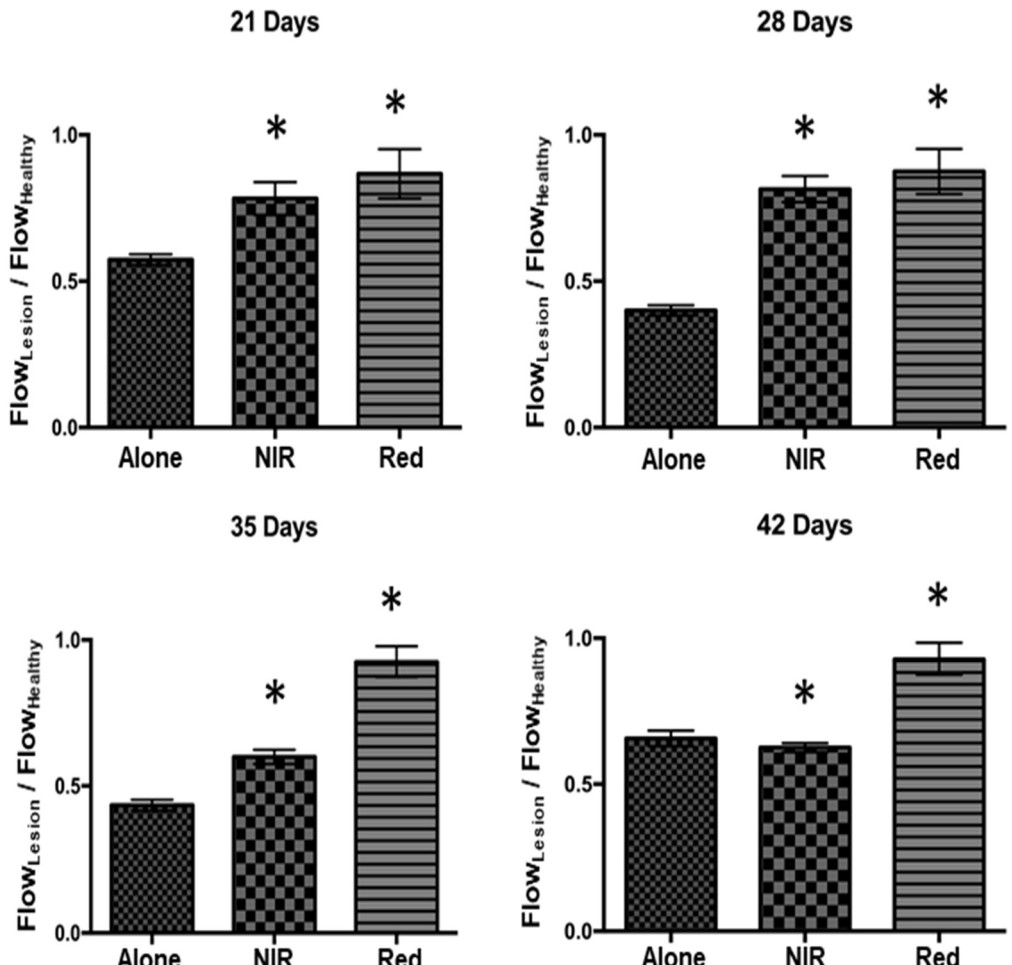

**Figure 3.** Vascular perfusion in radionecrosis lesion, compared with adjacent healthy skin in radiation and PBM-treated groups at 21, 28, 35, and 42 days. Data are presented as means with standard deviation; * denotes statistical significance $n = 6$, $p < 0.05$ with Kruskal–Wallis and Dunn's test.

### 3.3. Thermal Tissue Imaging for Inflammation

Another major effect of radiation-induced tissue damage is protracted inflammation. To examine this aspect in the radionecrosis lesions, we performed thermal imaging at 42 days, to correlate it with the increased vascular perfusion and improved cutaneous clinical presentation at 42 days (Figure 4). We observed a significant reduction in the inflammation in the PBM-treated groups.

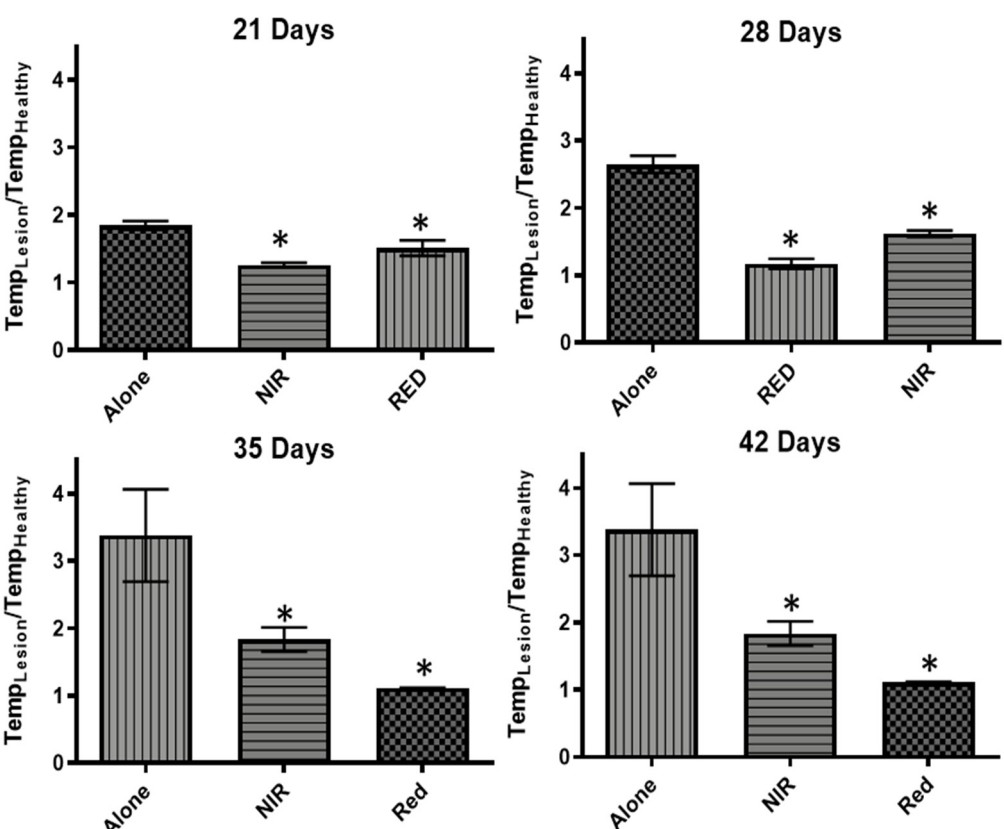

**Figure 4.** Thermal skin imaging to assess inflammation in radionecrosis lesion, compared with adjacent healthy skin in radiation and PBM-treated groups at 21, 28, 35, and 42 days. Data are presented as means with standard deviation; * denotes statistical significance $n = 6$, $p < 0.05$ with Kruskal–Wallis and Dunn's test.

### 3.4. Metabolic Tissue Analysis Using µPET-CT Imaging

A key aspect of radiation damage is the metabolic derangement of cells at the localized site, leading to tissue destruction and necrosis. To examine this aspect of radiation damage, we performed µPET-CT imaging with ${}^{18}$F-FDG at 42 days. We observed significant uptake and larger regions surrounding the ${}^{125}$I seed in the radiation group (Figure 5A). The NIR PBM-treated group demonstrated lower amounts and less accentuated uptake around the seed (Figure 5B), while the signal was least prominent in the red PBM-treated group (Figure 5C). These observations correlated with the increased thermal imaging, indicating inflammation and tissue damage observed in the radiation group alone was significantly attenuated by PBM treatments.

### 3.5. Histology Analysis

Following animal sacrifice, we validated the functional correlations of radionecrosis with histology. We performed H&E and Masson Trichrome staining to assess the tissues surrounding the ${}^{125}$I seed. We observed that radiation-exposed tissues demonstrated epithelial proliferation in the spinous zone (acanthosis) and prominent inflammatory infiltrate that included neutrophils and eosinophils (Figure 6). This was consistent with prior functional thermal imaging and µPET-CT analysis. The connective tissue also demonstrated large, coarse collagen with desmoplastic changes consistent with radiation-induced fibrosis evident clinically.

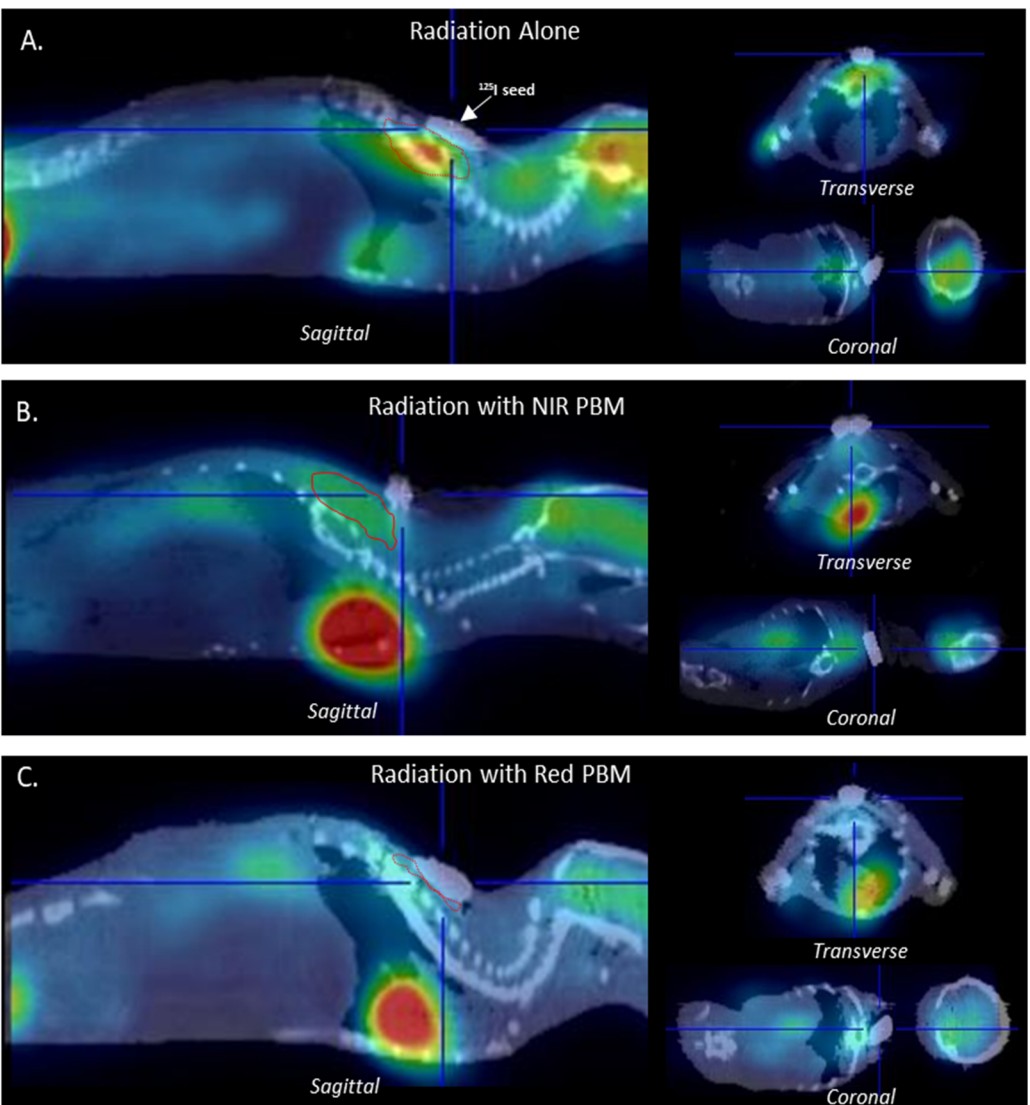

**Figure 5.** Fused images from CT and μPET to examine metabolic changes in radiation alone (**A**), NIR PBM treated radiation group (**B**), and red PBM treated radiation group (**C**). The [125]I seed is shown as a radiopaque, while the areas of [18]F-FDG uptake are outlined in areas below the seed with a red dotted line.

The PBM-treated groups had a relatively normal-looking appearance, with minimal epithelial changes in thickness or architecture and fewer inflammatory cells in the connective tissue. These changes were evident in H&E stained sections but more prominently highlighted by the Masson Trichrome staining.

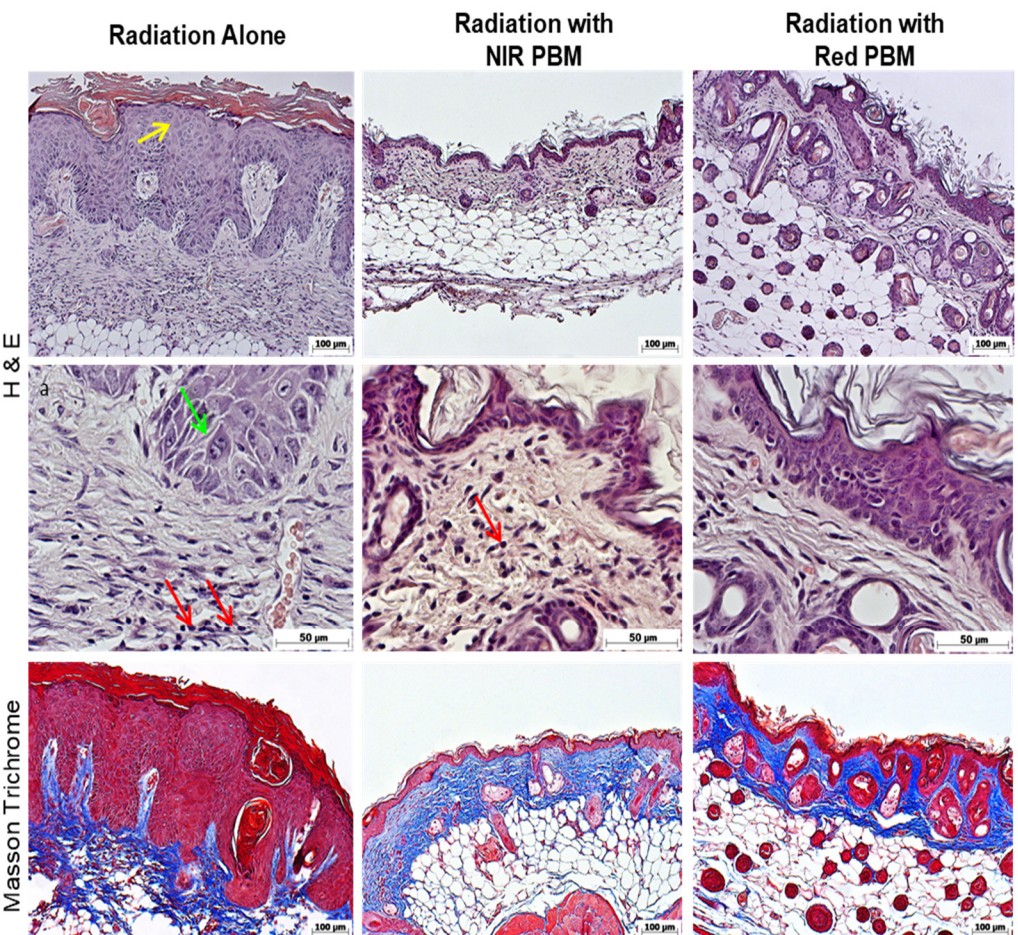

**Figure 6.** Histological analysis with H&E and Masson Trichrome staining of skin overlying the implanted [125]I seed at 42 days. The left panel shows tissues from the radiation alone, the middle depicts the radiation with NIR treatments, and the right panel shows radiation with red PBM-treated group. The yellow arrows highlight the hyperkeratotic regions with mitotic figures (green arrow), indicating cell proliferation in the spinous zone. The red arrows highlight the inflammatory cells infiltrating the connective tissue.

## 4. Discussion

PBM therapy has been known to be effective in visible to near-infrared wavelengths ranging from 400 to 1100 nm [26]. PBM studies comparing LED and lasers have noted that both devices are capable of similar therapeutic benefits such as reducing inflammation, increasing cell proliferation, stimulating angiogenesis, inducing granulation tissue formation, and increased synthesis of collagen and extracellular matrix in wound healing [26]. However, the direct benefits of PBM treatments in brachytherapy wounds have not been fully explored. The [125]I seed constantly emits radiation until the radioisotope is completely decayed. Although not so common, skin issues such as radiodermatitis and radionecrosis with brachytherapy remain challenging in some cases. The prominent inflammation, poor perfusion, and scarring with radionecrosis simulate burn wounds. Several novel treatments are being explored with burn wounds, including PBM treatments [27,28].

This study revealed that PBM treatments with red wavelength resulted in faster wound healing with dry desquamation surrounded by intense crusting at the borders by 42 days, with maximal severity at 28 days after the initial radionecrosis sign ~$1.7 \times 10^5$ Sv. In contrast, the NIR PBM-treated group had maximal severity at 35 days after the first radionecrosis sign ~$1.4 \times 10^5$ Sv, with healing evident at 49 days. These are 50% faster than the radiation-alone group that had maximal severity at 42 days ~$0.9 \times 10^5$ Sv. These results demonstrate the efficacy of both PBM wavelengths to effectively improve tissue resilience

and healing to radiation damage. These studies also observed improved efficacy of red wavelength over NIR for PBM therapy for this specific application. There could be two potential reasons for this observation. First, the more superficial targeting of PBM energy with red, compared with NIR, may enhance the cutaneous tissue responses around the subcutaneously implanted $^{125}$I seeds. Similar results have been observed with 810 nm PBM treatment for abrasion wounds that had a contraction and enhanced neo-vascularization, but discretely different results for re-epithelialization, compared with 635 nm PBM treatments and untreated controls [14]. Second, a novel concept in PBM dosimetry involves the inclusion of the individual biophotonics energy, termed quantum fluence [22]. This new dosimetry approach dictates different treatment times or irradiances must be employed for precise comparison. This remains to be examined in this brachytherapy-induced radionecrosis scenario in future studies.

We used a battery of functional assessments to objectively assess the efficacy of PBM in mitigating the radionecrosis damage in this study. These included noninvasive thermal tissue imaging, laser Doppler flowmetry, and µPET-CT imaging. Moreover, all changes were further validated with histological analysis. We observed reduced inflammation following PBM treatments, compared with radiation-alone groups with thermal imaging. This corroborated well with the histological analysis of reduced inflammatory infiltrate in these groups. However, a temperature difference of 1 °C can be used to detect angiogenesis, and such a range can be used also to detect benign tumors [29]. As vascular dysfunction leading to radionecrosis is a common sequela, we performed laser Doppler flowmetry and observed improved tissue vascular perfusion in PBM-treated groups. While the thermal imaging captures slight temperatures differences in the skin as signs of inflammation, it may also reflect the reduced blood flow and must be interpreted with caution. The LDF can detect microcirculatory changes due to cardiac pulsations, vasomotion, and the influence of the autonomic system on vascular tone [30]. Hence, both imaging enabled us to comprehensively analyze the therapeutic benefits of PBM in reducing local inflammation and improving perfusion in radionecrosis lesions. This study used a specific PBM dose to reduce the severity of radiation-induced damage and aid faster healing. Studies from our lab and other groups have shown that PBM dose demonstrates a limited range of reciprocity for irradiance and time [31–36]. While doses below this range are therapeutically ineffective, excessive doses can result in thermal damage and, at the very least, invalidate any therapeutic benefits.

µPET-CT with $^{18}$F-FDG is a functional imaging approach that provides unique molecular and metabolic information of tissues and organs based on glucose uptake capacity [30,37]. There remain some clinical concerns of biological risk of these combined PET/CT imaging, but it is still considered a powerful tool for clinical diagnostics [37]. Most tumors and inflammatory lesions lesion have increased uptake of $^{18}$F-FDG owing to enhanced glucose utilization [29,37]. More recently, several studies have examined the ability of $^{18}$F-FDG uPET-CT imaging to assess radiation damage [38,39]. This study noted a prominent increase in signal in the radiation-alone group that was dramatically reduced with PBM treatments. The precise cellular source of the $^{18}$F-FDG remains to be fully investigated, but it appears to correlate with the increased inflammatory infiltrate and radiation-induced adjacent tissues.

Given the increasing popularity of PBM treatments, especially in cancer patients with active tumor burden, as in brachytherapy, the effects on tumor cells remain an area of intense investigation. The ability to stimulate healing via cellular responses such as proliferation and migration has raised concerns on potential off-target effects on tumor cells. Several lines of evidence to date suggest that, while PBM has a modulatory effect on normal cellular responses, it appears to have an inhibitory response on tumor cells likely attributable to its inherently deranged metabolic and regulatory signaling [40]. These responses need to be carefully investigated further in well-designed labs and clinical  studies.

## 5. Conclusions

The side effects of ionizing radiation due to continued radiation emission on surrounding normal tissues lead to radiofibrosis and radionecrosis. PBM treatments can reduce these side effects by improving tissue resilience, thereby reducing incidences and promoting healing and resolution of lesions. This is the first report, to our knowledge, demonstrating the efficacy of PBM treatments in brachytherapy radiation wounds and warrants future investigations.

**Author Contributions:** Conceptualization, R.C.M., C.A.Z. and P.R.A.; methodology: R.C.M., C.A.Z. and P.R.A.; Software, R.C.M. and F.C.C.; LDF, G.E.C.N., S.N.S. and D.L.P.; Formal analysis, R.C.M. and P.R.A.; Investigation, R.C.M. and F.C.C.; LDF, G.E.C.N., S.N.S., D.L.P. and J.X.P.; Resources R.C.M., F.C.C.; LDF, G.E.C.N., S.N.S., D.L.P., J.X.P. and C.A.Z.; Data curation, R.C.M. and P.R.A.; Writing-original draft preparation, R.C.M.; Writing-review and editing, R.C.M. and P.R.A.; Visualization, R.C.M. and P.R.A.; Supervision, R.C.M. and P.R.A.; Project Administration, R.C.M. and P.R.A.; Funding acquisition, R.C.M. and C.A.Z. All authors have read and agreed to the published version of the manuscript.

**Funding:** This work was prepared with financial support from FAPESP through the fellowship grants #2014/18268-2 and #2016/22349-3.

**Institutional Review Board Statement:** The animal study protocol was approved by the Comite de Etica no uso de Animais (CEUA-IPEN) protocol code 160/15 on 08/13/15 for studies involving animals.

**Data Availability Statement:** All data were generated using GraphPad Prisma 7. Additionally, data are available upon request.

**Conflicts of Interest:** The authors declare no conflict of interest.

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
