# Peer review of "The Efficacy of Photobiomodulation Therapy in Improving Tissue Resilience and Healing of Radiation Skin Damage"

_photonics, doi:10.3390/photonics9010010_

Round 1

Reviewer 1 Report

This is a well-designed study regarding the efficacy of PBM therapy in improving tissue resilience and wound healing in the mice model treated by 125I brachytherapy. Minor points: In order to be a reproducible study, all of the PBM and device parameters for each wavelength should be reported; Manufacturer Model Identifier Year Produced Number & Type of Emitters (laser or LED) Wavelength [nm] Pulse mode [CW or Hz, duty cycle] Beam spot size at target [cm2] Irradiance at target [mW/cm2] If pulsed peak irradiance [mW/cm2] Exposure duration [sec] Fluence [J/cm2] Radiant energy [J] Number of points irradiated Area irradiated [cm2] Application technique Number and frequency of treatment sessions Total radiant energy over treatment course [J] In the methods section, provide details regarding the insertion of the radioisotope seeds subcutaneously, e.g., the exact anatomical location etc. Provide a reference supporting the effectiveness of the once-a-week application of PBM for wound healing in an animal model; specifically, brachytherapy and mice study. Because of the continuous nature of the brachytherapy irradiation, it seems that 3 times a week PBM could be more beneficial, this can be tested in future studies.

Author Response

Response to Reviewer comments

Reviewer 1

This is a well-designed study regarding the efficacy of PBM therapy in improving tissue resilience and wound healing in the mice model treated by 125I brachytherapy.

We thank the reviewer for their positive comments

Minor points: In order to be a reproducible study, all of the PBM and device parameters for each wavelength should be reported; Manufacturer Model Identifier Year Produced Number & Type of Emitters (laser or LED) Wavelength [nm] Pulse mode [CW or Hz, duty cycle] Beam spot size at target [cm2] Irradiance at target [mW/cm2] If pulsed peak irradiance [mW/cm2] Exposure duration [sec] Fluence [J/cm2] Radiant energy [J] Number of points irradiated Area irradiated [cm2] Application technique Number and frequency of treatment sessions Total radiant energy over treatment course [J]

Response: Thank you pointing out these methodological omissions, we have now added these details.

Changes made: Changes are highlighted on page 3, section 2.3.

In the methods section, provide details regarding the insertion of the radioisotope seeds subcutaneously, e.g., the exact anatomical location etc.

Response: Thank you pointing out these methodological omissions, we have now added these details. We would like to also point out the precise anatomical location is evident in Fig 5 (CT-PET).

Changes made: Changes are highlighted on page 3, section 2.2.

Provide a reference supporting the effectiveness of the once-a-week application of PBM for wound healing in an animal model; specifically, brachytherapy and mice study. Because of the continuous nature of the brachytherapy irradiation, it seems that 3 times a week PBM could be more beneficial, this can be tested in future studies.

Response: Thank you for your comment. We believe this is the very first report for PBM treatment for brachytherapy with Iodine seeds. We have previously performed various PBM treatments varying treatments per week, wavelengths, fluence, irradiance, application techniques in mice and rats that were published as conference abstracts at 2012 Sociedade Brasielra de Biosceincecias Nucleares (SBBN), World Federation at Laser Dentistry (WFLD) 2016, World Association for Photobiomodulation Therapy (WALT) 2016. Our data indicated PBM treatments once per week showed the best results and hence, was used in this study.

Changes made: We include the novelty of this study in the abstract on page 1 and note the choice of dosimetry on page 3, section 2.3 with a suitable citation [15].

Reviewer 2 Report

  1. In section 2.2, please explain the detail of the 6 groups studied.
  2. How was the fluence (20 J/cm2) being optimized? What is the fate of increased or decreased fluence ranges?
  3. Figures 2 and 4, the error bar shows only + values of SD and it must be indicating the ± values of SD. Please modify accordingly in the revision.
  4. Figure 2, please include the healing images for radiation alone (61 days) and radiation NIR-PBM (49 days) group treated models as indicated in Table 2.
  5. Please elaborate the section 2.8. The methodology must be given clearly, if you are using any standard protocols then please provide the reference.
  6. In section 2.8, the author mentioned that the “animals were sacrificed 42 days post-radiation and full-thickness skin biopsies were obtained from each group”. Please explain how the Table 2 study (Time course of radionecrosis lesion in animal study groups) was carried out?

Author Response

Response to Reviewer comments

Reviewer 2

  1. In section 2.2, please explain the detail of the 6 groups studied.

Response: Thank you for your comment, we have updated the description of the groups in methods section.

Changes made: Changes are highlighted on page 3, section 2.2.

  1. How was the fluence (20 J/cm2) being optimized? What is the fate of increased or decreased fluence ranges?

Response: Thank you for your question. To our knowledge, this is the first report for PBM treatment for brachytherapy with Iodine seeds. We have previously performed PBM treatments with varying parameters that were published as conference abstracts at 2012 Sociedade Brasielra de Biosceincecias Nucleares (SBBN), World Federation at Laser Dentistry (WFLD) 2016, World Association for Photobiomodulation Therapy (WALT) 2016. Our data indicated PBM treatments at 20J/cm2 once per week was most optimal. Hence, we used this protocol in this study.

In studies from our lab and several published studies in the literature, varying fluence either by changing irradiance or time can has limited reciprocity within a narrow range. However, fluence below the optimal range are ineffective while excessive fluence clearly generates thermal damage especially when using a laser PBM device.

Changes made: We now include these comments in the revised manuscript on page 10 with suitable citations [31-36].

  1. Figures 2 and 4, the error bar shows only + values of SD and it must be indicating the ± values of SD. Please modify accordingly in the revision.

Response: Thank you for pointing out these errors. These figures have been updated.

Changes made: Figures 2 and 4 have been updated on page 5 and 7 of the revised manuscript.

  1. Figure 2, please include the healing images for radiation alone (61 days) and radiation NIR-PBM (49 days) group treated models as indicated in Table 2.

Response: As suggested by thee reviewer, these images are now included in the ease find the images requested below.

  1. Please elaborate the section 2.8. The methodology must be given clearly, if you are using any standard protocols then please provide the reference.

Response: We regret this omission and have updated this section in the revised manuscript accordingly.

Changes made: The histology section has been revised on page 4, section 2.8.  

  1. In section 2.8, the author mentioned that the “animals were sacrificed 42 days post-radiation and full-thickness skin biopsies were obtained from each group”. Please explain how the Table 2 study (Time course of radionecrosis lesion in animal study groups) was carried out?

Response: We apologize for this typo. We have clarified our study outline in more detail as follows. Following implantation of the 125I seed, PBM treatments were begun weekly. The most several symptoms were evident at day 42 as noted in table 2. However, the study was continued till day 61 to complete the healing and other analysis. The maximal severity at 42 days was documented in our prior pilot studies and hence, one animal in each group was sacrificed for histological analyses.  

Changes made:  The 
